# Recurrent Laryngeal Nerve Preservation Strategies in Pediatric Thyroid Oncology: Continuous vs. Intermittent Nerve Monitoring

**DOI:** 10.3390/cancers13174333

**Published:** 2021-08-27

**Authors:** Rick Schneider, Andreas Machens, Carsten Sekulla, Kerstin Lorenz, Henning Dralle

**Affiliations:** 1Department of Visceral, Vascular and Endocrine Surgery, Martin Luther University Halle-Wittenberg, University Hospital, 06120 Halle, Germany; andreasmachens@aol.com (A.M.); carsten.sekulla@uk-halle.de (C.S.); kerstin.lorenz@uk-halle.de (K.L.); henning.dralle@uk-essen.de (H.D.); 2Section of Endocrine Surgery, Department of General, Visceral and Transplantation Surgery, University of Duisburg-Essen, 45147 Essen, Germany

**Keywords:** pediatric surgery, intraoperative nerve monitoring, loss of signal, recurrent laryngeal nerve, vocal cord palsy

## Abstract

**Simple Summary:**

Thyroid operations in children are difficult because children have thinner nerves than adults, and there is less space for the surgeon to operate. Since it runs closely behind the thyroid capsule, the nerve innervating the vocal cords can be injured during the operation. In thyroid cancer, the thyroid gland typically needs to be removed completely, putting the nerve at greater risk of injury. This surgical risk can be reduced by monitoring the function of the nerve before it is lastingly damaged. There are two methods to achieve this: intermittent (longer intervals between pulses) and continuous (very small intervals between pulses) nerve stimulation. In this study of 258 children with suspected or confirmed thyroid cancer, nerve damage and vocal cord palsy were observed only after intermittent and not after continuous nerve stimulation. This demonstrated that continuous nerve stimulation was safer than intermittent nerve stimulation.

**Abstract:**

(1) Background: Pediatric thyroidectomy is characterized by considerable space constraints, thinner nerves, a large thymus, and enlarged neck nodes, compromising surgical exposure. Given these challenges, risk-reduction surgery is of paramount importance in children, and even more so in pediatric thyroid oncology. (2) Methods: Children aged ≤18 years who underwent thyroidectomy with or without central node dissection for suspected or proven thyroid cancer were evaluated regarding suitability of intermittent vs. continuous intraoperative neuromonitoring (IONM) for prevention of postoperative vocal cord palsy. (3) Results: There were 258 children for analysis, 170 girls and 88 boys, with 486 recurrent laryngeal nerves at risk (NAR). Altogether, loss of signal occurred in 2.9% (14 NAR), resulting in six early postoperative vocal cord palsies, one of which became permanent. Loss of signal (3.5 vs. 0%), early (1.5 vs. 0%), and permanent (0.3 vs. 0%) postoperative vocal cord palsies occurred exclusively with intermittent IONM. With continuous nerve stimulation, sensitivity, specificity, positive and negative predictive values, and accuracy reached 100% for prediction of early and permanent postoperative vocal cord palsy. With intermittent nerve stimulation, sensitivity, specificity, positive and negative predictive values, and accuracy were consistently lower for prediction of early and permanent postoperative vocal cord palsy, ranging from 78.6% to 99.8%, and much lower (54.2–57.9%) for sensitivity. (4) Conclusions: Within the limitations of the study, continuous IONM, which is feasible in children ≥3 years, was superior to intermittent IONM in preventing early and permanent postoperative vocal cord palsy.

## 1. Introduction

The operative morbidity of pediatric thyroid surgery is well defined for various age groups of children who undergo thyroidectomy at a tertiary referral center for Graves’ disease [1], papillary and medullary thyroid cancer [2], and neoplastic C-cell disease associated with multiple endocrine neoplasia type 2 [3]. Pediatric thyroidectomy is characterized by considerable space constraints, thinner nerves, a large thymus, and enlarged neck nodes, compromising surgical exposure.

In pediatric thyroid oncology, removal of all thyroid cancer from the neck frequently requires extensive dissection around the recurrent laryngeal nerve, increasing the risk of injury when embarked on by an inexperienced surgeon [4]. Breathing and swallowing greatly impact on health-related quality of life, which, if impaired, become a pain point for affected children and their parents [5,6]. Given these challenges, surgical risk reduction is of paramount importance in children. This is even more important when surgery is the mainstay of therapy, as in thyroid cancer. For risk minimization, the American Association of Clinical Endocrinology (AACE) and the American Head and Neck Society Endocrine Surgery Section (AHNS) jointly advocate intraoperative neuromonitoring (IONM) [7].

Recently, continuous IONM was shown to measure nerve electrophysiology more accurately than intermittent IONM during thyroidectomy for mostly benign thyroid conditions in children, regardless of age [6]. No such data has been put forward for children with oncological thyroid conditions. The present investigation aimed to evaluate the performance of intermittent vs. continuous IONM on early and permanent postoperative vocal cord palsy in this high-risk group of children.

## 2. Methods

### 2.1. Study Design

Included in this comparative study were all children aged ≤18 years who underwent thyroidectomy with or without central node dissection for suspected or proven thyroid cancer between May 1998 and April 2021 at the authors’ institution. Written informed consent was obtained for all procedures, which represented the standard of care in Germany [8].

A total of 4 consultant surgeons, each with an annual surgical volume of more than 150 thyroid procedures, operated on the children following the same standard approach. All procedures were carried out using 2.5–3.5-fold optical magnification and conventional bipolar diathermy. Intermittent IONM, routinely employed from the beginning of the study, was gradually superseded by continuous IONM in 2011, and largely phased out after 2012, filling a niche when continuous IONM equipment was unsuited for infants and small children or refused to work properly [6]. All children underwent intraoperative neuromonitoring with standardized stimulation of the vagus nerve and recurrent laryngeal nerve before and after resection [9].

Each child had preoperative laryngeal exams, typically within several weeks of surgery, and postoperative laryngeal exams 2 days after surgery [10]. Abnormal postoperative laryngeal exams were repeated regularly until they became normal. Postoperative vocal cord palsies failing to resolve within 6 months were considered as permanent.

For retrospective analysis of existing data sets from routine clinical care, no institutional review board approval was required under national law and applicable institutional regulations.

### 2.2. Setting of the Neuromonitoring Equipment

The setting of intermittent and continuous IONM equipment, described in more depth elsewhere [5,6,11,12,13], can be briefly summarized as follows:

For intermittent IONM, standard handheld monopolar stimulation probes, the Neurosign 100™ system (Inomed GmbH, Emmendingen, Germany), and the NIM-response^®^ 2.0 and NIM-response^®^ 3.0 neuromonitoring systems (Medtronic, Jacksonville, FL, USA) were used (4.0 Hz, 100 µs, 1 mA). Beginning in October 2003, electromyographic (EMG) signals generated by stimulation of the ipsilateral vagal nerve were recorded via tube electrodes for quantitative analysis.

For continuous IONM, the Automatic Periodic Stimulation (APS^®^) circumferential clip electrode was employed together with a standard handheld monopolar stimulation probe (Medtronic, Jacksonville, FL, USA) (4.0 Hz, 100 µs, 1 mA), the NIM-response^®^ 3.0 with a pulse generator for continuous stimulation (1.0 Hz, 100 µs, 1 mA), and an EMG amplifier.

Stable evoked potentials from the vocal muscles were registered via endotracheal tube surface electrodes (NIM Standard Reinforced EMG Endotracheal Tube and NIM Contact Reinforced EMG Endotracheal Tube, Medtronic, Jacksonville, FL, USA). If necessary, the anesthetist was asked to reposition the tracheal tube to maximize the signal amplitude to at least 500 μV.

### 2.3. Statistical Analysis

All clinical and electrophysiological data were prospectively registered using Microsoft^®^ SQL Server 2016 (Redmond, Washington, DC, USA).

For statistical analysis, the software package SPSS^®^ version 25.0 (IBM, Armonk, New York, NY, USA) was used. Categorical data were tested with the two-tailed chi-square test, and are presented as absolute numbers with percentages. If the test reached the significance level of *p* < 0.05, correction for multiple testing was performed according to Bonferroni [14]. Continuous data were compared with the two-tailed Mann–Whitney–Wilcoxon rank sum test, and are displayed as median (interquartile range).

Sensitivity, specificity, positive predictive value (PPV), negative predictive value (NPV), and accuracy were calculated as follows: Sensitivity equals predicted vocal cord palsies divided by the sum of true positive and false negative EMG signals. Specificity equals true negative results divided by the sum of true negative and false positive results. PPV equals true positive results divided by the sum of true positive and false positive results. NPV equals true negative results divided by the sum of true negative and false negative results. Accuracy equals the sum of true positive and true negative results divided by all tests performed.

## 3. Results

### 3.1. Demographics, Thyroid Histopathology, Extent, Type, and Completeness of Pediatric Oncological Thyroidectomy by Types of Intraoperative Nerve Stimulation and Recording Electrodes

There were 258 consecutive children for analysis, 170 girls and 88 boys, with 486 recurrent laryngeal nerves at risk (NAR).

When grouped by type of intraoperative nerve stimulation (Table 1), the 210 patients with intermittent stimulation (404 NAR) were, by and large, comparable to the 48 patients with continuous stimulation (82 NAR), barring few exceptions. After correction for multiple testing, significant differences remained regarding disease spectrum (decrease of hereditary medullary thyroid cancer from 33.8 to 8.3%), primary tumor diameter (medians of 1.3 vs. 2.5 mm), and concomitant use of tube electrodes (40.5 vs. 100%), reflecting temporal trends. Temporal shifts in the age distribution in the use of intermittent versus continuous IONM between January 1998 and April 2021 are shown in Figure 1.

When grouped by type of recording electrodes (Table 2), the 125 children with needle electrodes (244 NAR) differed in many ways from the 133 children with tube electrodes (242 NAR), most of which reflected temporal trends in terms of age at thyroidectomy (medians of 12.0 vs. 14.0 years; fewer children ≤6 years and more children >12 years had tube electrodes), disease spectrum (decrease of hereditary medullary thyroid cancer from 43.2 to 15.8% and increase of papillary thyroid cancer from 34.4 to 66.2%), primary tumor diameter (medians of 0.4 vs. 2.1 mm), extent of neck surgery (increase of less than total thyroidectomies with central node dissection from 16.8 to 34.6%, paralleled by reduction of total thyroidectomies with central node dissection from 66.4 to 46.6%), and concomitant use of continuous nerve stimulation (0 vs. 36.1%).

All in all, there were one postoperative hemorrhage (0.8%) and one wound infection (0.8%), each of which necessitated reoperation (data not shown).

### 3.2. Intraoperative Loss of the EMG Signal and Postoperative Vocal Cord Palsy in Children with Normal Preoperative Vocal Cord Function

A total of 486 NAR were intraoperatively monitored, 14 of which (2.9%) revealed loss of the EMG signal. This resulted in six early unilateral postoperative nerve palsies, one of which became permanent.

When grouped by type of intraoperative nerve stimulation (Table 3, upper panel), only children with intermittent nerve stimulation experienced loss of the EMG signal (3.5 vs. 0%) and early (1.5 vs. 0%) and permanent (0.3 vs. 0%) postoperative vocal cord palsies, with loss of signal trending towards statistical significance (*p* = 0.087). Temporal shifts in the number of intact vocal cord function and early postoperative vocal fold palsies in the use of intermittent versus continuous IONM between January 1998 and April 2021 are shown in Figure 2.

When grouped by type of recording electrodes (Table 3, lower panel), children with needle electrodes sustained loss of the EMG signal (3.3 vs. 2.5%) and early (1.6 vs. 0.8%) and permanent (0.4 vs. 0%) postoperative vocal cord palsies slightly more often than children with tube electrodes.

### 3.3. Characterization of Children with Postoperative Vocal Cord Palsies

Loss of the EMG signal and postoperative vocal cord palsies were seen with intermittent nerve stimulation only.

Altogether, four early postoperative vocal cord palsies (three on the left and one on the right) occurred after primary operation for hereditary C-cell disease, one of which became permanent (Table 4). Conversely, two early postoperative vocal cord palsies (one on either side) were noted after completion of neck surgery for hereditary medullary and follicular thyroid cancer, respectively.

Taken together, early vocal cord palsy affected 4 (2.3%) of 172 children ≤10 years of age (among all 208 children), as compared with 2 (0.9%) of 232 children >10 years of age (among all 278 children) with intermittent nerve stimulation, but this association was nonsignificant (*p* = 0.234; *p* = 0.229) for all children.

It was interesting to note that it was the child with the most advanced tumor (extrathyroid extension, 23 dissected node metastases, and microscopically involved surgical margins) who developed the only permanent postoperative vocal cord palsy.

### 3.4. Prediction of Early and Permanent Postoperative Vocal Cord Palsy by Types of Intraoperative Nerve Stimulation and Recording Electrodes

For technical evaluation of the performance of IONM, all recurrent laryngeal nerves were considered, whether preoperatively functioning or not or subsequently resected for oncological reasons. Due to this, 26 (instead of 6) early and 21 (instead of 1) permanent postoperative vocal cord palsies were available for evaluation.

With continuous nerve stimulation, sensitivity, specificity, positive and negative predictive values, and accuracy reached 100% for prediction of early and permanent postoperative vocal cord palsy (Table 5, left column).

With intermittent nerve stimulation, sensitivity, specificity, positive and negative predictive values, and accuracy were consistently lower for prediction of early and permanent postoperative vocal cord palsy, ranging from 78.6% to 99.8%, and much lower (54.2–57.9%) for sensitivity (Table 5, right column).

In the comparative analysis of the recording tubes, sensitivity, specificity, and positive and negative predictive values for prediction of early and permanent postoperative vocal cord palsy were variable, without a clear pattern favoring one electrode type over the other (Table 6).

## 4. Discussion

This is the largest and most comprehensive study and first of its kind to compare intermittent with continuous monitoring of recurrent laryngeal nerve function in pediatric thyroid surgical oncology. The present work was noteworthy for the large number of infants and small children who underwent neck surgery for hereditary C-cell disease. Remarkably, not a single instance of loss of signal, or early or permanent postoperative vocal fold palsy, was noted (based on 82 nerves at risk) during or after continuous nerve monitoring, as opposed to intermittent nerve monitoring.

These results were consistent with the findings of a recent study of 6029 mostly adult patients with mainly benign thyroid disease in which continuous IONM independently reduced early postoperative vocal cord palsy 1.8-fold and permanent postoperative vocal cord palsy 29.4-fold compared with intermittent IONM [13]. This study found 1 permanent vocal cord palsy per 75.0 early postoperative vocal cord palsies with continuous IONM (based on 5208 nerves at risk), as compared with 1 permanent vocal cord palsy per 4.2 early postoperative vocal cord palsies after intermittent IONM (based on 5024 nerves at risk). The present findings support the notion that these adult data are applicable to children with pediatric thyroid oncology as well.

Impending recurrent laryngeal nerve injury is preceded by combined events in the EMG tracing, which combine amplitude decreases >50% with latency increases >10% relative to baseline (Figure 3) [11]. Most nerve injuries are caused by traction on the recurrent laryngeal nerve, which is commonly reversible upon immediate release of the nerve [15,16]. Real-time detection of nerve injuries as they unfold enables prompt reversal of causative maneuvers during the operation. This competitive advantage is the unique selling point of continuous IONM in experienced hands, giving it a lead over intermittent IONM [13].

Previous research suggested that this competitive edge applies to children as well [6]. When continuous nerve stimulation was used, median baseline amplitudes augmented with the child’s age. This was likely due to better device connectivity (reduced resistance between electrodes and effector) in older children, in whom the mismatch in size between the APS^®^ electrode and the vagus nerve is diminished. Likewise, median latency became larger with increasing age of the child, and thus the length of the monitored vagus/recurrent laryngeal nerve axis, which increase was greater on the left side where the recurrent laryngeal nerve takes a longer course beneath the aortic arch [6].

The present data, deriving from a high-risk group of children who underwent surgery for thyroid cancer and hereditary C-cell disease, confirm and extend these findings.

For children aged <10 years, the American Thyroid Association management guidelines for children with thyroid nodules and differentiated thyroid cancer advocate consideration of the use of IONM, despite the lack of pertinent data [17]. Subsequently, it was reported that the risk of postoperative vocal fold palsy does not increase in children of younger age groups when pediatric thyroidectomy was performed in infants and young children by experienced high-volume surgeons [1,2]. These findings may not equally extend to other clinical settings [18]. By way of contrast, a recent study of 113 patients with 183 nerves at risk found that the risk of nerve injury was increased in children <10 years of age, especially in the context of thyroid cancer and concomitant central neck dissection [19]. In the present study, four of our six patients who developed early postoperative vocal cord palsy were aged ≤10 years, but this difference was not statistically significant (Table 4). 

The relatively large size of commercially available endotracheal tubes with integrated reporting electrodes, featuring inner diameters of ≥5.0 mm, is an obstacle to the use of IONM in children under 3 years of age (Figure 4). For very young children under the age of 3 years, the use of needle electrodes can be a viable option. Needle electrodes, if displaced, are quickly spotted and easily repositioned because they are located inside the operating field [20]. Adhesive electrodes can also be fitted onto generic endotracheal 3.0 or 4.0 tubes under the premise that utmost caution is exercised to avoid a mismatch between cuff and tip of the endotracheal tube. Such mismatch increases the risk of injury to tracheal mucosa and wall, and may jeopardize maintenance of the pediatric airway.

Congruent with other research in the field, this study was limited by the use of two risk-minimization strategies at the same time, visual nerve identification and IONM. Both strategies jointly represent a standard of care in Germany [13]. As a result, the independent contribution of each method cannot be separated.

When a novel technology is introduced into clinical practice, there typically is a learning curve that may affect results. As an institutional standard, intermittent IONM was well established, whereas continuous IONM was still an emerging technology at its introduction at the authors’ institution in 2011. However, it was this novel technology, not the established standard of care, that was associated with lower, not higher, rates of early postoperative and permanent vocal cord palsy.

Further limitations, inherent in research into a rare disease, included the long study period and the low event rate of postoperative vocal cord palsy at the authors’ referral center. Although levels of institutional standardization and staff continuity (four consultant surgeons in total) were unusually high, subtle improvements in surgical skills over the study period cannot be entirely excluded.

To add another layer of complexity, children do not really form a homogeneous group of people. Biologically, it may be worthwhile to subdivide children into newborns, infants, preschool children, school children, and adolescents. This line of reasoning is buttressed by the difficulty of fitting children aged <3 years with commercial endotracheal tubes with integrated reporting electrodes necessitated the use of needle electrodes for intermittent nerve stimulation in the early years of the study. Younger children also tend to have their thymus preserved more often than older children, exacerbating surgical space constraints.

However, when it comes to subdividing a small group of people with infrequent disease and rare clinical outcomes, statistical issues immediately come to the fore. In fact, pediatric research is almost always hampered by low numbers of study participants and the infrequency of clinical outcomes, diminishing statistical power and hindering evidence generation for children.

Hereditable C-cell disease also is more frequent in young children, whereas papillary and follicular thyroid cancers preferentially affect older children. In addition, thyroid cancer encompasses a wide disease spectrum, ranging from node-negative tumors (C-cell disease and follicular thyroid cancer) to regionally advanced node-positive tumors (papillary and medullary thyroid cancer). By implication, extent of disease and extent of node dissection were required to match up. The independent effects of these interdependencies between age and thyroid disease are difficult, if not impossible, to unravel.

A key strength of this investigation was the high level of standardization with rigorous control of preoperative and postoperative vocal fold function. However, the present findings were obtained at a high-volume tertiary referral center for pediatric and adult oncologic surgery. Outside specialist centers, the operative morbidity is likely to be considerably greater.

## 5. Conclusions

Within the limitations of this observational study, continuous IONM, which is feasible in children ≥3 years of age, was superior to intermittent IONM in preventing postoperative vocal cord palsy in pediatric thyroid oncology. When combined with the use of handheld stimulation probes to enhance nerve identification, continuous IONM is poised to become an integral part of pediatric thyroid cancer surgery. The authors of the present research use 3.5-fold optical magnification for children <3 years of age, and 2.5-fold optical magnification for older children to locate nerves, as well as the parathyroid glands, which are small, translucent, and not easy to distinguish in these children from adjacent tissues, thymus, and central neck nodes. Before terminating neck operations, the authors stimulate the vagus nerve one more time to confirm that the recurrent laryngeal nerve is working, and perform parathyroid hormone quick tests to determine upfront the adequacy of postoperative parathyroid function, which is paramount to infants and small children. In light of this, it is prudent to have high-volume surgeons and a multidisciplinary team manage these children [6,18,21,22].

## Figures and Tables

**Figure 1 cancers-13-04333-f001:**
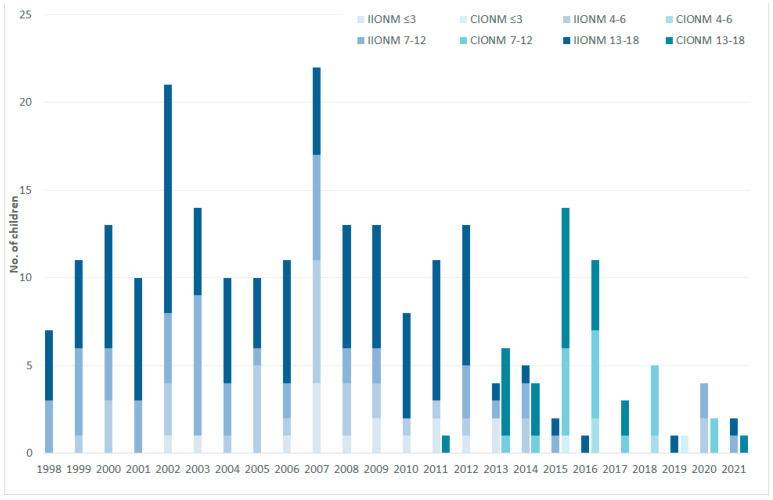
Age distribution of thyroid surgery per year by intermittent vs. continuous neuromonitoring (IIONM vs. CIONM), January 1998–April 2021.

**Figure 2 cancers-13-04333-f002:**
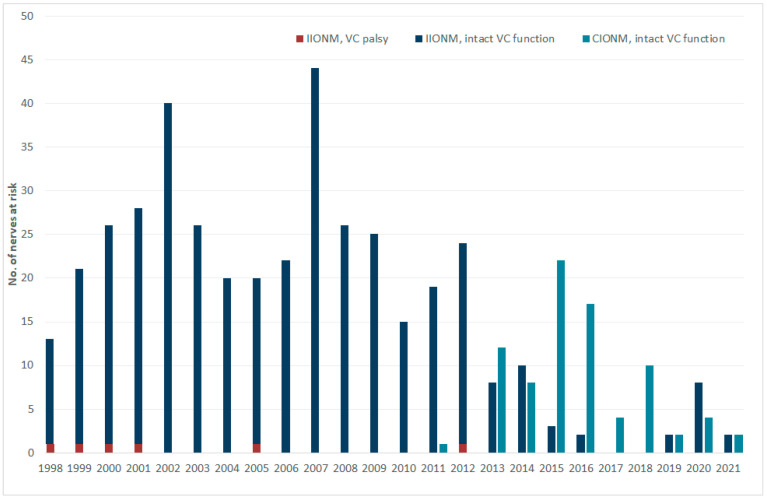
Number of intact vocal cord functions and vocal fold palsies after thyroid surgery per year by intermittent vs. continuous neuromonitoring (IIONM vs. CIONM), January 1998–April 2021.

**Figure 3 cancers-13-04333-f003:**
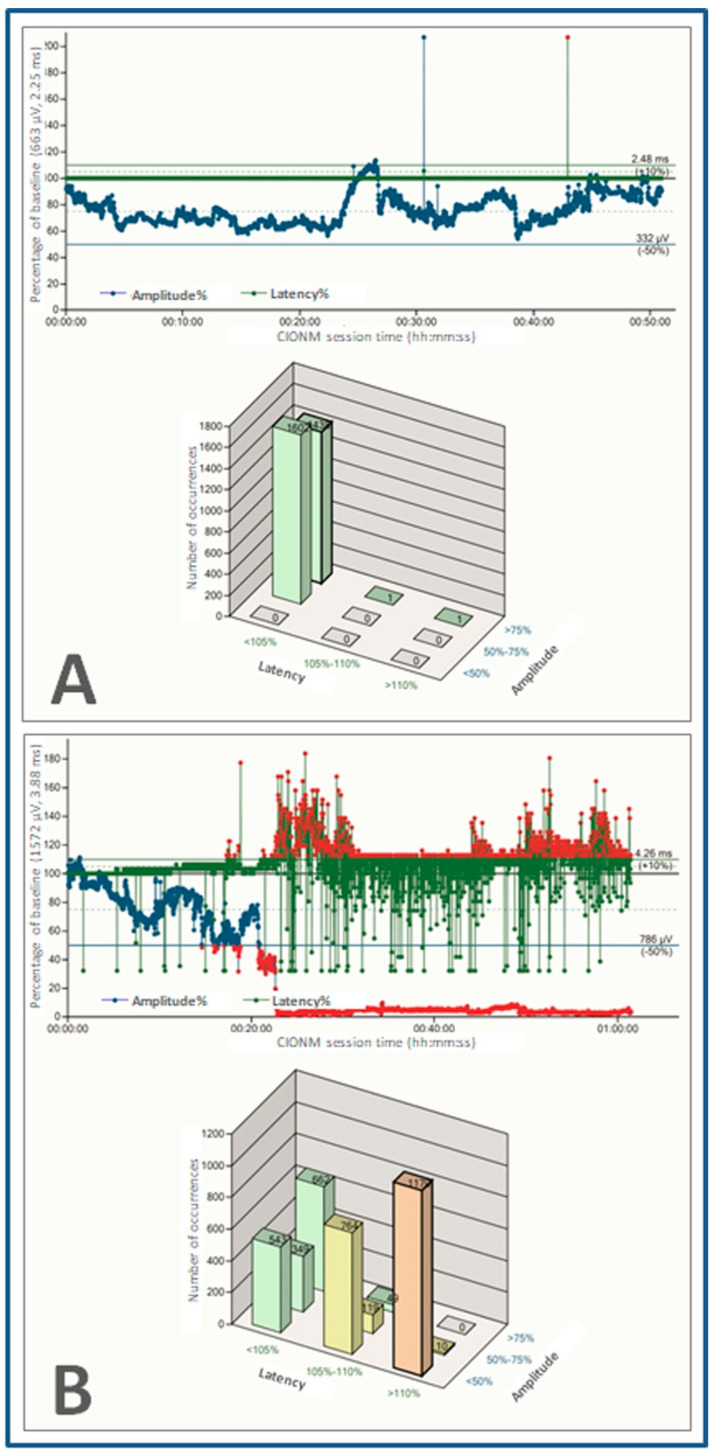
Electromyogram tracings with preset amplitude (blue line) and latency (dark green line) thresholds with automatic periodic stimulation at 1 Hz. (**A**) Normal electromyogram (amplitude of 663 µV and latency of 2.25 ms at baseline) heralding normal postoperative vocal cord function. (**B**) Definitive loss of signal (combined event) without intraoperative recovery, signaling an ipsilateral vocal cord palsy.

**Figure 4 cancers-13-04333-f004:**
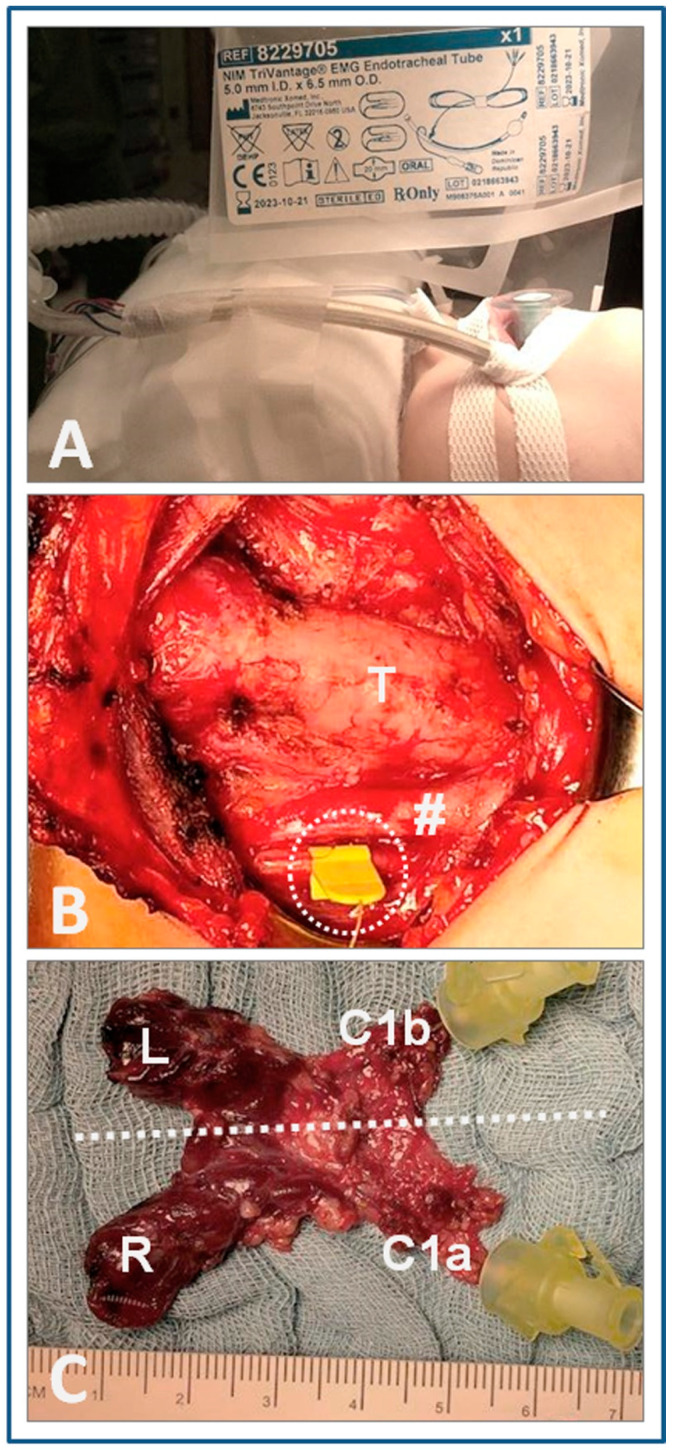
Continuous intraoperative nerve stimulation in a 3.5-year-old girl with C-cell hyperplasia in the context of multiple endocrine neoplasia type 2A. (**A**) Fitted recording endotracheal tube surface electrodes (NIM Trivantage EMG Endotracheal Tube 5.0). (**B**) Intraoperative situs after complete mobilization and removal of thyroid gland and central neck nodes, with the monopolar APS^®^ clip electrode riding on the right vagus. (**C**) Picture of the contiguous surgical specimen comprising both thyroid lobes and central node compartment. The circle identifies the monopolar APS^®^ clip electrode; the hashtag the right common carotid artery; and the dashed line the midline; whereas T signifies the trachea; L/R the left/right thyroid lobe; and C1a/C1b the right/left central node compartment.

**Table 1 cancers-13-04333-t001:** Demographics, thyroid histopathology, extent, type, and completeness of pediatric oncological thyroidectomy by type of intraoperative nerve stimulation.

Variables	Total (258 Children; 486 Nerves at Risk)		Intermittent Nerve Stimulation (210 Children; 404 Nerves at Risk)		Continuous Nerve Stimulation (48 Children; 82 Nerves at Risk)		*p*
Sex, male	88	34.1	69	32.9	19	39.6	0.375
Age at operation, years, median (range)	13.0	(7.0; 16.0)	13.0	(7.0; 16.0)	12.5	(9.0; 15.0)	0.765
Age at operation, years							
≤3	18	7.0	16	7.6	2	4.2	0.086
4–6	36	14.0	33	15.7	3	6.3	
7–12	70	27.1	51	24.3	19	39.6	
13–18	134	51.9	110	52.4	24	50.0	
Type of neoplastic condition							
Hereditary C-cell hyperplasia	35	13.6	24	11.4	11	22.9	0.036
Medullary thyroid cancer	75	29.1	71	33.8	4	8.3	<0.001 *
Papillary thyroid cancer	131	50.8	103	49.0	28	58.3	0.246
Follicular thyroid cancer	12	4.7	8	3.8	4	8.3	0.179
Thyroid metastases from other cancers	5	1.9	4	1.9	1	2.1	0.935
Largest primary tumor diameter, mm, median (range) (based on 153 children)	1.6	(0.3; 2.6)	1.3	(0.2; 2.5)	2.5	(1.5; 3.9)	<0.001 *
Extent of neck surgery							
Less than total thyroidectomy without central neck dissection	8	3.1	7	3.3	1	2.1	0.160
Less than total thyroidectomy with central neck dissection	67	26.0	55	26.2	12	25.0	
Total thyroidectomy without central neck dissection	38	14.7	26	12.4	12	25.0	
Total thyroidectomy with central neck dissection	145	56.2	122	58.1	23	47.9	
Type of surgery							
Primary surgery	170	65.9	132	62.9	38	79.2	0.032
Reoperation	88	34.1	78	37.1	10	20.8	
Resectional status (based on 163 children)							
Complete (R0)	151	92.6	118	91.5	33	97.1	0.376
Microscopically incomplete (R1)	8	4.9	7	5.4	1	2.9	
Grossly incomplete (R2)	4	2.5	4	3.1	0	0	
Number of cleared nodes on central neck dissection, median (range) (based on 212 children)	7	(1; 16)	8	(2; 16)	6	(0; 13)	0.125
Number of cleared nodes on central neck dissection (based on 212 children)							
≤5	53	25.0	45	25.4	8	22.9	0.622
6–10	51	24.1	40	22.6	11	31.4	
11–15	33	15.6	27	15.3	6	17.1	
16–20	28	13.2	25	14.1	3	8.6	
>20	47	22.2	40	22.6	7	20.0	
Type of recording electrode used							
Needle electrode	125	48.4	125	59.5	0	0	<0.001 *
Tube electrode	133	51.6	85	40.5	48	100	

* After Bonferroni correction for multiple testing.

**Table 2 cancers-13-04333-t002:** Demographics, thyroid histopathology, extent, type, and completeness of pediatric oncological thyroidectomy by type of recording electrodes.

Variables	Total (258 Children; 486 Nerves at Risk)		Needle Electrode (125 Children; 244 Nerves at Risk)		Tube Electrode (133 Children; 242 Nerves at Risk)		*p*
Sex, male	88	34.1	41	32.8	47	35.3	0.667
Age at operation, years, median (range)	13.0	(7.0; 16.0)	12.0	(6.0; 15.0)	14.0	(10.0; 16.0)	<0.001 *
Age at operation, years							
≤3	18	7.0	12	9.6	6	4.5	0.001 *
4–6	36	14.0	24	19.2	12	9.0	
7–12	70	27.1	36	28.8	34	25.6	
13–18	134	51.9	53	42.4	81	60.9	
Type of neoplastic condition							
Hereditary C-cell hyperplasia	35	13.6	20	16.0	15	11.3	0.268
Medullary thyroid cancer	75	29.1	54	43.2	21	15.8	<0.001 *
Papillary thyroid cancer	131	50.8	43	34.4	88	66.2	<0.001 *
Follicular thyroid cancer	12	4.7	6	4.8	6	4.5	0.912
Thyroid metastases from other cancers	5	1.9	2	1.6	3	2.3	0.703
Largest primary tumor diameter, mm, median (range) (based on 153 children)	1.6	(0.3; 2.6)	0.4	(0.2; 2.0)	2.1	(1.0; 3.2)	<0.001 *
Extent of neck surgery							
Less than total thyroidectomy without central neck dissection	8	3.1	4	3.2	4	3.0	0.990
Less than total thyroidectomy with central neck dissection	67	26.0	21	16.8	46	34.6	<0.001 *
Total thyroidectomy without central neck dissection	38	14.7	17	13.6	21	15.8	0.768
Total thyroidectomy with central neck dissection	145	56.2	83	66.4	62	46.6	<0.001 *
Type of surgery							
Primary surgery	170	65.9	90	72.0	80	60.2	0.045
Reoperation	88	34.1	35	28.0	53	39.8	
Resectional status (based on 163 children)							
Complete (R0)	151	92.6	68	94.4	83	91.2	0.259
Microscopically incomplete (R1)	8	4.9	3	4.2	5	5.5	
Grossly incomplete (R2)	4	2.5	1	1.4	3	3.3	
Number of cleared nodes on central neck dissection, median (range) (based on 212 children)	7	(1; 16)	7	(2; 16)	8	(1; 17)	0.815
Number of cleared nodes on central neck dissection, number (based on 212 children)							
≤5	53	25.0	28	26.9	25	23.1	0.098
6–10	51	24.1	25	24.0	26	24.1	
11–15	33	15.6	15	14.4	18	16.7	
16–20	28	13.2	19	18.3	9	8.3	
>20	47	22.2	17	16.3	30	27.8	
Type of nerve stimulation used							
Intermittent	210	81.4	125	100	85	63.9	<0.001 *
Continuous	48	18.6	0	0	48	36.1	

* After Bonferroni correction for multiple testing.

**Table 3 cancers-13-04333-t003:** Intraoperative loss of the EMG signal and postoperative vocal cord palsy in children with normal preoperative vocal cord function.

A. Type of Intraoperative Nerve Stimulation							
Variables	Total (486 Nerves at Risk)		Intermittent (404 Nerves at Risk)		Continuous (82 Nerves at Risk		*p*
Loss of the EMG signal	14	2.9	14	3.5	0	0	0.087
Early postoperative vocal cord palsy	6	1.2	6	1.5	0	0	0.267
Permanent postoperative vocal cord palsy	1	0.2	1	0.3	0	0	0.652
**B. Type of Recording Electrodes**							
**Variables**	**Total** **(486 Nerves** **at Risk)**		**Needle Electrodes** **(244 Nerves** **at Risk)**		**Tube** **Electrodes** **(242 Nerves** **at Risk)**		***p***
Loss of the EMG signal	14	2.9	8	3.3	6	2.5	0.598
Early postoperative vocal cord palsy	6	1.2	4	1.6	2	0.8	0.417
Permanent postoperative vocal cord palsy	1	0.2	1	0.4	0	0	0.319

EMG, electromyogram.

**Table 4 cancers-13-04333-t004:** Children with normal preoperative nerve function and early postoperative vocal cord palsy.

Child		Thyroid Histopathology	Neck Surgery			Intraoperative Nerve Stimulation		Postoperative Vocal Cord Palsy		
Age at TT *	Sex	Type of Neoplastic Condition (AJCC/TNM Classification)	Extent	No. of Involved Nodes	Resectional Status	Type	Loss of EMG Signal	Side	Type	Outcome
3	male	hereditary MTC/MEN 2B (pT1aN0M0)	TT, CND, LND	0 of 91	Complete (R0)	intermittent	none	left	incomplete	transient
4	male	hereditary CCH/MEN2A	TT, CND	0 of 2	Complete (R0)	intermittent	none	right	in-complete	transient
7	male	hereditary MTC/MEN2A (pT1aN1bM0)	TT, CND, LND	2 of 42	Complete (R0)	intermittent	none	left	complete	transient
10	female	hereditary MTC/MEN2A (T4bN1bM1)	LND, MND for completion	23 of 41	Microscopically incomplete (R1)	intermittent	type 2	left	complete	permanent
16	female	hereditary MTC/MEN2A (pT1aN0M0)	TT, CND	0 of 8	Complete (R0)	intermittent	type 1	left	complete	transient
17	female	FTC (pT3aN0M0)	TT, CND for completion	0 of 19	Complete (R0)	intermittent	type 2	left	complete	transient

CCH, C-cell hyperplasia; CND, central neck dissection; EMG, electromyogram; FTC, follicular thyroid cancer; LND, lateral neck dissection; MTC, medullary thyroid cancer; MND, modified neck dissection; type 1, segmental loss of signal; type 2, global loss of signal; TT, total thyroidectomy. * 4 of 172 nerves at risk in children ≤10 years of age vs. 2 of 232 nerves at risk in children >10 years of age with intermittent nerve stimulation (*p* = 0.234).

**Table 5 cancers-13-04333-t005:** Prediction of early and permanent postoperative vocal cord palsy by type of intraoperative nerve stimulation.

	Type of Intraoperative Nerve Stimulation					
Early Postoperative Vocal Cord Palsy	Intermittent			Continuous		
	EMG Signal Per Nerve *			EMG Signal Per Nerve **		
	Normal	Abnormal	Total	Normal	Abnormal	Total
Absent	397	1	398	82	0	82
Present	11	13	24	0	2	2
Total	408	14	422	82	2	84
		%	95%CI	%	95%CI	P
Sensitivity		54.2	(34.2, 74.1)	100	(100, 100)	0.621
Specificity		99.8	(99.3, 100.2)	100	(100, 100)	0.836
PPV		92.9	(79.4, 106.3)	100	(100, 100)	0.750
NPV		97.3	(95.7, 98.9)	100	(100, 100)	0.494
Accuracy		97.2	(95.5, 98.8)	100	(100, 100)	0.118
**Permanent Postoperative Vocal Cord Palsy**	**Intermittent**			**Continuous**		
	**EMG Signal Per Nerve ***			**EMG Signal Per Nerve ****		
	**Normal**	**Abnormal**	**Total**	**Normal**	**Abnormal**	**Total**
Absent	400	3	403	82	0	82
Present	8	11	19	0	2	2
Total	408	14	422	82	2	84
		%	95%CI	%	95%CI	P
Sensitivity		57.9	(35.7, 80.1)	100	(100, 100)	0.244
Specificity		99.3	(98.4, 100.1)	100	(100, 100)	0.433
PPV		78.6	(57.1, 100.1)	100	(100, 100)	0.646
NPV		98.0	(96.7, 99.4)	100	(100, 100)	0.563
Accuracy		97.4	(95.9, 98.9)	100	(100, 100)	0.134

EMG, electromyogram; NPV, negative predictive value; PPV, positive predictive value. * Including preoperative vocal cord palsies and oncological recurrent laryngeal nerve resections to increase the number of outcome events from 6 early and 1 permanent to 24 early and 19 permanent postoperative vocal cord palsies. ** Including preoperative vocal cord palsies and oncological recurrent laryngeal nerve resections to increase the number of outcome events from 0 early and 0 permanent to 2 early and 2 permanent postoperative vocal cord palsies.

**Table 6 cancers-13-04333-t006:** Prediction of early and permanent postoperative vocal cord palsy by type of recording electrodes.

	Type of Recording Electrodes					
Early Postoperative Vocal Cord Palsy	Needle Electrodes			Tube Electrodes		
	EMG Signal Per Nerve *			EMG Signal Per Nerve **		
	Normal	Abnormal	Total	Normal	Abnormal	Total
Absent	240	0	240	239	1	240
Present	8	8	16	5	5	10
Total	248	8	256	244	6	250
		%	95%CI	%	95%CI	P
Sensitivity		50.0	(25.5, 74.5)	50.0	(19.0, 81.0)	>0.999
Specificity		100.0	(0, 100.0)	99.6	(98.8, 100.4)	0.316
PPV		100.0	(100.0, 100.0)	83.3	53.5, 113.2)	0.259
NPV		96.8	(94.6, 99.0)	98.0	(96.2, 99.7)	0.415
Accuracy		96.9	(94.7, 99.0)	97.6	(95.7, 99.5)	0.619
**Permanent Postoperative** **Vocal Cord Palsy**	**Needle Electrodes**			**Tube Electrodes**		
	**EMG Signal Per Nerve ***			**EMG Signal Per Nerve ****		
	**Normal**	**Abnormal**	**Total**	**Normal**	**Abnormal**	**Total**
Absent	242	1	243	240	2	242
Present	6	7	13	4	4	8
Total	248	8	256	244	6	250
		%	95%CI	%	95%CI	P
Sensitivity		53.9	(26.7, 80.9)	50.0	(15.4, 84.6)	0.864
Specificity		99.6	(98.8, 100.4)	99.2	(98.0, 100.3)	0.560
PPV		87.5	(64.6, 110.4)	66.7	(28.9, 104.4)	0.375
NPV		97.6	(95.7, 99.5)	98.4	(96.8, 100.0)	0.538
Accuracy		97.3	(95.2, 99.3)	97.6	(95.7, 99.5)	0.812

EMG, electromyogram; NPV, negative predictive value; PPV, positive predictive value. * Including preoperative vocal cord palsies and oncological recurrent laryngeal nerve resections to increase the number of outcome events from 4 early and 1 permanent to 16 early and 13 permanent postoperative vocal cord palsies. ** Including preoperative vocal cord palsies and oncological recurrent laryngeal nerve resections to increase the number of outcome events from 2 early and 0 permanent to 10 early and 8 permanent postoperative vocal cord palsies.

## Data Availability

Data are not available.

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
