# Peer review of "Recurrent Laryngeal Nerve Preservation Strategies in Pediatric Thyroid Oncology: Continuous vs. Intermittent Nerve Monitoring"

_cancers, 2021, doi:10.3390/cancers13174333_

Round 1

Reviewer 1 Report

In this study, R. Schneider et al. have compared continuous vs intermittent nerve monitoring in pediatric thyroid oncology. They investigated 258 children with thyroid cancer who underwent thyroid operations and found that continuous is safer than intermittent nerve stimulation in relation to nerve damage and vocal cord palsy, though not statistically significant. They also compare needle versus tube electrodes for monitoring.

The manuscript is well written and provides important new data to the field.

However, what is missing is a discussion about the long time period where the data was collected (1998-2021):

  • Intermittent intraoperative neuromonitoring was seldom carried out after 2012. Did other surgical techniques improve during this long time course, i.e. did surgeons get better, so that not that not just the change from intermittent to continuous monitoring improved the outcome.
  • Postoperative vocal cord palsy was only observed in the intermittent group (6 out of 404 nerves at risk) vs 0 out of 82 nerves at risk). A difference, which was not statistically significant. When were these palsies observed, during the whole time period or just at the beginning?

Minor comments:

  • In line 69, I would remove the word ‘unsurprisingly’.
  • In lines 124-25, the authors write that they use Oracle and Microsoft to register data. However, they do not write exactly which software.
  • Lines 287-89 should be moved to the figure legend for figure 2.

Author Response

Dear Reviewer,

We greatly appreciate the thoughtful review and would like to thank you for you valuable suggestions on how to improve our manuscript further.

Please find our point-to-point response:

  1. “Intermittent intraoperative neuromonitoring was seldom carried out after 2012. Did other surgical techniques improve during this long time course, i.e. did surgeons get better, so that not that not just the change from intermittent to continuous monitoring improved the outcome.”

This important aspect now is addressed in the discussion section (page 15 of 18):

“When a novel technology is introduced into clinical practice, there typically is a learning curve that may affect results. As institutional standard, intermittent IONM was well established, whereas continuous IONM was still an emerging technology at its introduction at the authors’ institution in 2011.

Although levels of institutional standardization and staff continuity (4 consultant surgeons in total) were unusually high, subtle improvements in surgical skills over the study period cannot be entirely excluded.”

  1. “Postoperative vocal cord palsy was only observed in the intermittent group (6 out of 404 nerves at risk) vs 0 out of 82 nerves at risk). A difference, which was not statistically significant. When were these palsies observed, during the whole time period or just at the beginning?”

This point is well taken. Information on the development of vocal cord palsy by type of nerve monitoring over time now is detailed in the new Figure 2 (page 10 of 18).

Figure 2 revealed that vocal fold palsies in the intermittent group were somewhat more common towards the earlier phase of the study.

The issue of the surgical learning curve now is discussed as a limitation of the study (page 15 of 18):

When a novel technology is introduced into clinical practice, there typically is a learning curve that may affect results. As institutional standard, intermittent IONM was well established, whereas continuous IONM was still an emerging technology at its introduction at the authors’ institution in 2011. Yet it was this novel technology, not the established standard of care, that was associated with lower, not higher, rates of early postoperative and permanent vocal cord palsy.”

  1. “In line 69, I would remove the word ‘unsurprisingly’.”

In response, the word ‘unsurprisingly has been removed as suggested (page 4 of 18)

Unsurprisingly, bBreathing and swallowing greatly impact on health-related quality of life, which, if impaired, becomes a pain point for affected children and their parents [5,6].”

  1. “In lines 124-25, the authors write that they use Oracle and Microsoft to register data. However, they do not write exactly which software.”

This missing detail has been included in the methods section (page 5 of 18):

“All clinical and electrophysiological data were prospectively registered using Microsoft® SQL Server 2016Oracle® (Redmond, Washington, USARedwood City, California, USA) and Microsoft® (Redmond, Washington, USA).”

  1. “Lines 287-89 should be moved to the figure legend for figure 2.”

Thank you very much for kindly pointing out this formatting error of ours in the legend to Figure 2, which has been amended as suggested (page 15 of 18).

Best regards,

Rick

Reviewer 2 Report

The present paper is interesting, well conducted and well written, I have no objections. I think it deserves to be published in the current form.

Author Response

Dear Reviewer,

We greatly appreciate the thoughtful review and would like to thank you for you valuable feedback to our manuscript.

Best regards,

Rick

Reviewer 3 Report

In this study, the authors evaluated the preventive effects of nerve monitoring for RLN in thyroid cancer surgery in children. The authors found that continuous monitoring is the effective way for preventing RLN. The result of this study is in the common sense physiologically as the continuous monitoring might give the more chance of prevention for RLN. As the thyroid cancer of children is rare, this small number of included participants were livewithable.

<Major>

The biggest problem is the allocation of patients between two group (intermittent Vs continuous). The comparison between them is not fair in the current status.

First of all, the included participants of continuous monitoring were only 1/4 of intermittent. The authors did not find any complication in the continuous monitoring group. This might be continuous monitoring is superior way than intermittent. However, it could be just from the low number of continuous monitoring group. Despite I agree with authors conclusion of this study, the scientific conclusion should be proved without the suspicion of coincidence. At least this should have been proved by statistically. On the other hand, the P-value was 0.087/0.267/0.652. In the current status, the author cannot insist the continuous monitoring is superior than intermittent monitoring.

The authors should give the information who did this surgery. How many surgeons were involved in this study? Actually, I think the experience of surgeon’s skill is the most important factor for the prevent of RLN injury in the thyroid cancer.

I think the periods of using the intermittent and continuous monitoring might be different. This makes the possibility of difference of surgeon skill. Especially, this study was performed in the single hospital from 1998 through 2021, the ability of hospital facility, and the level of surgeon’s skill might be very different. Additionally, I guess the continuous monitoring was performed in the recent periods, this is not a negligible factor. The authors should give the information of periods of included participants of each two group. In the article of surgical methods, this bias commonly appeared. The methods used posteriorly commonly have better results than previous methods not because of technical superiority, but because of just time of study.

The included participants are not just children. The authors included participants until 18 years old. The body size of adolescent is quite not different from the adults. Even though the authors give the information of age group in table 1, I suggest the authors give the information of the height category additionally.

As the number of vocal cord palsy is 0 in the continuous monitoring group, the data of sensitivity, specificity is not give the reader enough information when judging the use of nerve monitoring. This information is not useful in the current results. Additionally, the information of sensitivity, specificity is commonly useful for the diagnosis of disease rather than selecting the operation methods.

<Minor>

Discussion, the first paragraph, why C-cell disease was described specifically, while this study includes other types of thyroid cancer?

Author Response

Dear Reviewer,

We greatly appreciate the thoughtful review and would like to thank you for you valuable suggestions on how to improve our manuscript further.

Please find our point-to-point response:

  1. “.... The authors did not find any complication in the continuous monitoring group. This might be because continuous monitoring is superior to intermittent.

However, it could be just from the low number of continuous monitoring group. Despite that, I agree with authors’ conclusion of this study.

The scientific conclusion should be proved without the suspicion of coincidence. At least this should have been proved statistically. On the other hand, the P-value was 0.087/0.267/0.652.

In the current status, the author cannot insist the continuous monitoring is superior than intermittent monitoring.”

This point is well taken.

For clarity, a new Figures 2 has been added to better illustrate this important point.

Moreover, the discussion has been considerably expanded to better put the present findings into perspective (page 15 of 18):

“When a novel technology is introduced into clinical practice, there typically is a learning curve that may affect results. As institutional standard, intermittent IONM was well established, whereas continuous IONM was still an emerging technology at its introduction at the authors’ institution in 2011. Yet it was this novel technology, not the established standard of care, that was associated with lower, not higher, rates of early postoperative and permanent vocal cord palsy.

Further limitations, inherent in research into a rare disease, included the long study period and the low event rate of postoperative vocal cord palsy at the authors’ referral center. Although levels of institutional standardization and staff continuity (4 consultant surgeons in total) were unusually high, subtle improvements in surgical skills over the study period cannot be entirely excluded.”

  1. “How many surgeons were involved in this study? … possibility of difference of surgeon skill.

Especially, this study was performed in the single hospital from 1998 through 2021, the ability of hospital facility, and the level of surgeon’s skill might have been very different.”

This is a possibility indeed.

More information about the surgical skills of the operating surgeons now is provided in the methods (page 4 of 18) and discussion (15 of 18) sections:

“A total of 4 consultant surgeons, each with an annual surgical volume of more than 150 thyroid procedures, operated on the children following the same standard approach.”

“Although levels of institutional standardization and staff continuity (4 consultant surgeons in total) were unusually high, subtle improvements in surgical skills over the study period cannot be entirely excluded.”

  1. “… the periods of using the intermittent and continuous monitoring might be different. … continuous monitoring was performed more often in the recent period, this is not a negligible factor.

The authors should give the information of periods of included participants of each two group. The methods used more recently commonly have better results than previous methods not because of technical superiority, but because of just time of study.”

This missing information now is graphically provided in two new figures (Figure 1 and Figure 2), which have been added to the manuscript (pages 7 of 18 and 10 of 18).

The issue of the surgical learning curve now is discussed as a limitation of the study (page 15 of 18):

“When a novel technology is introduced into clinical practice, there typically is a learning curve that may affect results. As institutional standard, intermittent IONM was well established, whereas continuous IONM was still an emerging technology at its introduction at the authors’ institution in 2011. Yet it was this novel technology, not the established standard of care, that was associated with lower, not higher, rates of early postoperative and permanent vocal cord palsy.

Further limitations, inherent in research into a rare disease, included the long study period and the low event rate of postoperative vocal cord palsy at the authors’ referral center. Although levels of institutional standardization and staff continuity (4 consultant surgeons in total) were unusually high, subtle improvements in surgical skills over the study period cannot be entirely excluded.”

  1. “The authors included participants until 18 years old. The body size of adolescent is quite not different from the adults. Even though the authors give the information of age group in table 1, I suggest the authors give the information of the height category additionally.”

The age distribution of the study population indeed is an important aspect.

Child height, unlike child age, was not routinely recorded in the patients’ charts.

To depict the age distribution of the children operated over the years under intermittent vs. continuous nerve monitoring, a new Figure 1 has been added to the manuscript (page 7 of 18).

Furthermore, the heterogeneity of children is specifically addressed as a limitation of the study (page 16 of 18):

“To add another layer of complexity, children do not really form a homogeneous group of people. Biologically, it may be worthwhile to subdivide children into newborns, pre-school children, school children and adolescents. This line of reasoning is buttressed by the difficulty of fitting children aged <3 years with commercial endotracheal tubes that have recording electrodes integrated. This dilemma necessitated the use of needle electrodes for intermittent nerve stimulation in the early years of the study. Younger children also tend to have their thymus preserved more often than older children, exacerbating surgical space constraints. 

However, when it comes to subdividing a small group of people with infrequent disease and rare clinical outcomes, statistical issues immediately come to the fore. In fact, pediatric research is almost always hampered by low numbers of study participants and the infrequency of clinical outcomes, diminishing statistical power and hindering evidence generation for children.”

  1. “As the number of vocal cord palsy is 0 in the continuous monitoring group, the data of sensitivity, specificity do not give the reader enough information when judging the use of nerve monitoring … Additionally, the information of sensitivity, specificity is commonly useful for the diagnosis of disease rather than selecting the operation methods.”

Sensitivity and specificity of a technology (e.g., intraoperative nerve monitoring), quantifying the capability of the technology to predict clinical outcomes (e.g., postoperative vocal cord palsy), are highly informative variables for surgeons.

The caveat “for technical evaluation” is made upfront in results (page 11 of 18):

“For technical evaluation of the performance of IONM, all recurrent laryngeal nerves were considered, whether preoperatively functioning or not or subsequently resected for oncological reasons.”

  1. “Discussion, the first paragraph, why C-cell disease was described specifically, while this study includes other types of thyroid cancer?”

Surgically, there should be no difference between performing thyroidectomy for neoplastic C-cell hyperplasia or thyroidectomy for medullary thyroid cancer confined to the thyroid gland.

Eliminating children who postoperatively were found to harbor C-cell disease only from the study would have shifted the study population towards older children (page 16 of 18):

“Hereditable C-cell disease also is more frequent in young children, whereas papillary and follicular thyroid cancers preferentially affect older children. Besides, thyroid cancer encompassed a wide disease spectrum, ranging from node-negative tumors (C-cell disease and follicular thyroid cancer) to regionally advanced node-positive tumors (papillary and medullary thyroid cancer). By implication, extent of disease and extent of node dissection needed to match up. The independent effects of such these interdependencies between age and thyroid disease are hard, if not impossible, to unravel.”

Best regards,

Rick

Round 2

Reviewer 3 Report

I think this study have the problem in the nature of study design. To prove usefulness of the operation skill, it must be compared in fairly. As this study was not designed previously, this study just shows the results of single hospital. 

Even though I agree that the continuous monitoring might be useful compared to intermittent monitoring, this comparison is compared in the current status. 

I think the results of this study should be read carefully admitting the basic limitation of  this study design.